# Porcelain Insulator Crack Location and Surface States Pattern Recognition Based on Hyperspectral Technology

**DOI:** 10.3390/e23040486

**Published:** 2021-04-20

**Authors:** Yiming Zhao, Jing Yan, Yanxin Wang, Qianzhen Jing, Tingliang Liu

**Affiliations:** State Key Laboratory of Electrical Insulation and Power Equipment, Xi’an Jiaotong University, Xi’an 710049, China; yiming_zhao96@163.com (Y.Z.); yxwang199501@163.com (Y.W.); qzjing199901@163.com (Q.J.); liu1741770229@163.com (T.L.)

**Keywords:** hyperspectral imaging technology, canny edge detection, EfficientNet, pattern recognition

## Abstract

A porcelain insulator is an important part to ensure that the insulation requirements of power equipment can be met. Under the influence of their structure, porcelain insulators are prone to mechanical damage and cracks, which will reduce their insulation performance. After a long-term operation, crack expansion will eventually lead to breakdown and safety hazards. Therefore, it is of great significance to detect insulator cracks to ensure the safe and reliable operation of a power grid. However, most traditional methods of insulator crack detection involve offline detection or contact measurement, which is not conducive to the online monitoring of equipment. Hyperspectral imaging technology is a noncontact detection technology containing three-dimensional (3D) spatial spectral information, whereby the data provide more information and the measuring method has a higher safety than electric detection methods. Therefore, a model of positioning and state classification of porcelain insulators based on hyperspectral technology is proposed. In this model, image data were used to extract edges to locate cracks, and spectral information was used to classify the surface states of porcelain insulators with EfficientNet. Lastly, crack extraction was realized, and the recognition accuracy of cracks and normal states was 96.9%. Through an analysis of the results, it is proven that the crack detection method of a porcelain insulator based on hyperspectral technology is an effective non-contact online monitoring approach, which has broad application prospects in the era of the Internet of Things with the rapid development of electric power.

## 1. Introduction

Insulators, especially porcelain insulation, play an important role in ensuring that electrical equipment meets insulation requirements. However, due to their structure, porcelain insulators are vulnerable to mechanical damage, especially in the process of vibration due to external forces such as earthquakes. Thus, the external porcelain insulator structure is easily damaged and produces cracks, thus reducing the insulation performance of insulators. If the insulator damage is not detected in time, its long-term operation is accompanied by the expansion of cracks [1], whereby the insulation performance of the insulator is reduced, resulting in a safety hazard.

Compared with surface contamination of porcelain insulators, the requirements and specifications for surface cracks of porcelain insulators are more stringent. China follows the GB/T 772-2005 specification (Chinese National Standard) for porcelain insulators, in which there are certain requirements for various defects of porcelain insulators that according to the different operating conditions have different requirements: for electrical and distribution devices with porcelain, the porcelain is not allowed to crack; porcelains that are used as main insulation and bear large impact are allowed to have cracks on the surface of the umbrella edge 10 mm away from the main part. Other porcelain parts are allowed to have cracks on the surface beyond 10 mm from the electrode site. The width of the above cracks shall not exceed 0.05 mm, the single length shall not exceed 10 mm, and the total length of the cracks shall not exceed (total surface defect area /5) mm. The main defects of porcelain insulators are predominantly caused by vibration and external forces, most of which occur in the root area of porcelain bushings, leading to cracks. The insulator itself exhibits slight changes due to the manufacturing process and other factors, whereas earthquakes can aggravate the crack expansion. At present, there are many crack detection methods for porcelain insulators, including manual inspection, acoustic fault detection, ultrasonic detection, X-ray detection, and leakage current detection.

The method of manual inspection can identify obvious foreign objects or damage; however, for small breakages and cracks, detection efficiency and accuracy are difficult to guarantee. Kyu-Chil Park et al. used the sound fault detection method to classify fault state insulators by taking into account the total noise and the number of 120 Hz harmonic components [2,3]; however, this method is restricted by the distance used and the accuracy of the detection. Accordingly, it cannot be applied to online testing, and it is mostly used in the production of insulators. The laser detection method is more accurate in finding the location and size of cracks [4]; however, offline inspection of the device is also required. Accordingly, online detection cannot be carried out. Shen and He Wei used the leakage current method to detect the presence of broken insulators by detecting the electrical characteristics of porcelain insulator parameters [5,6], but this method cannot be used to locate the cracks or defects, and the measured results are unstable due to the influence of weather and insulator structural parameters. In addition, there are many applications of images to detect insulators; for example, Tao et al. used aerial photographic images to detect the beam failure of an insulator [7], while Zhai et al. used aerial images combined with Convolutional Neural Network (CNN) to detect the integrity of an insulator [8]. However, this method is more effective for the detection of severely damaged insulators, whereas the detection of small cracks has certain limitations.

Therefore, in order to realize online, contactless, and more accurate detection of porcelain insulator defects, as well as reduce the power equipment failure caused by insulator defects, this paper proposes a porcelain insulator defect detection method based on hyperspectral technology. As a new image detection technology, hyperspectral nondestructive detection has been applied in agricultural production, environmental and cultural relic protection, and other applications, showing good results. For example, the hyperspectral method was used to detect cracks in paintings to quantify cultural heritage [9,10]. Zhang et al. applied this approach to crop defect detection, as well as other quality and safety detection methods [11,12,13,14]. Wang et al. applied the hyperspectral method in the field of geographical remote sensing terrain classification [15,16,17,18,19]. The advantage of the hyperspectral detection method is that it can realize noncontact online monitoring during the operation of the equipment, it can effectively find tiny cracks and defects that are difficult to distinguish using the naked eye, and it can effectively evaluate the status of porcelain insulators. With the development of online detection in power systems, there will be higher requirements for detection means, and hyperspectral detection method can meet the requirements of detection due to its characteristics. A large amount of data can be retained in the online monitoring process, which provides important data support for artificial intelligence methods. At that time, deep learning and other artificial intelligence methods will be more widely used in the fault diagnosis of power equipment.

The main contributions of this paper are as follows:

(1) A state diagnosis method for porcelain insulators based on hyperspectral technology is proposed. By using the image data and spectral data in hyperspectral data, the surface crack of the porcelain insulator was located, and the surface state was recognized. The online contactless detection of porcelain insulators was effectively realized, which is an effective supplement to the existing insulator detection methods.

(2) A hyperspectral detection platform was established. The hyperspectral test platform was built with the hyperspectral camera as the main one. Through black and white correction, multiple scattering correction, and spectral dimension reduction processing, the dimensionality disaster problem of spectral data was solved, and the data was processed into simple and usable data, which provided a database for the subsequent pattern recognition of porcelain insulator crack state.

(3) In this article, through EfficientNet network data the spectrum pattern recognition, EfficientNet within prescribed parameters and calculation of the limit to the expansion of the multidimensional network, effectively improves the accuracy of pattern recognition, and compared with the traditional machine learning, saves artificial feature extraction step, reducing the artificial extraction characteristics that affect the results.

## 2. Proposed Method

### 2.1. Fundamental Theory of Hyperspectral Imaging

After one-dimensional information in space passes through the lens and slits, different wavelengths of light are generated according to different degrees of dispersion transmission through the grating to form a band, with radiation hitting the detector. The detector, thus, receives each pixel’s position according to the intensity of the indicative spectrum. A detector array composed of multiple detectors is used to receive spectral information, as shown in Figure 1a. 

Schematics of hyperspectral imaging are shown in Figure 1b. In the process of imaging, each pixel corresponds to a spectral segment, and a line corresponds to a spectral surface. Therefore, the detector receives a spatial line of spectral information at each time point before obtaining a spatial two-dimensional image through mechanical push and sweep. This allows for completing the image of the entire plane, along with spectral data acquisition, thereby obtaining a data cube.

### 2.2. Crack Localization Based on Computer Image Vision

Image diagnosis based on computer vision technology can detect and locate insulator cracks, according to the angle of an image. First, the main recognition process generally involves dealing with the noise of an image, followed by the image’s gray level. Then, Canny edge detection is applied to connect the cracks with morphology [20,21] before extracting the skeleton from each connected domain to get complete crack forms.

#### 2.2.1. Image Denoising

First of all, in the process of image acquisition, although the experiment has tried to simulate the natural light, the image is still likely to produce Gaussian noise due to uneven illumination. Gaussian noise obeys Gaussian distribution, which is one of the most widely common types of noise among all and appears more in RGB images and images with uneven illumination. Therefore, in the processing of this paper, the noise is mainly Gaussian noise, which is one of the reasons for choosing the Gaussian filter.

Secondly, there are many methods of image denoising, including median filtering, mean filtering, box filtering, Gaussian filtering, and so on. The Gaussian filter is linear, and the coefficient in the convolution template decreases with the increase of the distance from the template center. Compared with the mean filter, the Gaussian filter has less blur on the whole image, which can effectively suppress noise and smooth the image. Several denoising methods are compared in this paper, as shown in Figure 2, a picture of a circuit breaker is used here as an example.

The Gaussian filter shows better performance among other noise filters; thus, it was selected for noise removal. The generation equation of a Gaussian filter kernel of size (2k+1)×(2k+1) is given as follows: (1)$$H_{ij}=\frac{1}{2\pi \sigma^{2}}\exp\left(-\frac{{\left(i-\left(k+1\right)\right)}^{2}+{\left(j-\left(k+1\right)\right)}^{2}}{2\sigma^{2}}\right);1\leq i,j\leq \left(2k+1\right),$$
where $H_{ij}$ is the generated Gaussian filter.
(2)$$e=H*A=\left[\begin{array}{ccc} h_{11} & \cdots & h_{1j}\\ \vdots & \ddots & \vdots \\ h_{i1} & \cdots & h_{ij} \end{array}\right]*\left[\begin{array}{ccc} a_{11} & \cdots & a_{1n}\\ \vdots & \ddots & \vdots \\ a_{m1} & \cdots & a_{mn} \end{array}\right],$$
where A is the original image data matrix, which is convolved with the Gaussian filter to obtain the denoising matrix e. The denoising results are shown in Figure 3a.

#### 2.2.2. Grayscale Gradient Calculation

The most important feature of the edge is that the gray value changes dramatically. If the gray value is regarded as a binary function value, then the change in gray value can be described by a gradient. A pixel has eight neighborhoods; therefore, the Canny algorithm uses four calculators to detect the horizontal, vertical, and diagonal edges in an image. 

In the gray calculation, this paper considers a variety of commonly used methods, such as Robert operator, Laplacian of Gaussian (LoG) filter method, Sobel operator, and the Prewitt operator. The Robert operator positioning is more accurate, but because it does not include smoothing, it is more sensitive to noise. The LoG filter method determines the edge point by detecting the second derivative zero-crossing. The problem of this approach is that there is a contradiction between the precision of edge positioning and the level of noise elimination. Therefore, the noise level and the precision of edge positioning should be appropriately selected according to the specific problems. Prewitt operator and Sobel operator are the first-order differential operators, and both of them have better detection effect on images with gray gradient and low noise, the former is average filtering, while the latter is weighted average filtering, and the edge of the detected image may be greater than 2 pixels, so this paper chooses the Sobel operator for gray calculation. 

The Sobel calculator is used here to calculate the difference between the two-dimensional image on the *x*-axis and the *y*-axis. The two templates are obtained from the original graph, and a differential graph of the *x*-axis and *y*-axis is drawn, which allows calculating the gradient G and direction θ of this point. The result of image processing is shown in Figure 3b.
(3)$$G=\sqrt{G_{x}^{2}+G_{y}^{2}}$$
(4)$$\theta =\arctan\left(G_{y}^{}/G_{x}^{}\right)$$

#### 2.2.3. Non-Maximum Suppression (NMS) to Control the Edge Width

The edges detected by the Sobel operator are relatively thick; thus, it is necessary to suppress the pixel points with an insufficient gradient and only retain those with the maximum gradient, thereby eliminating a large number of points. An edge that is multiple pixels wide can be changed to one that is single-pixel wide to achieve a thin edge. Non-maximum suppression is an important step in edge detection. In a popular sense, it refers to looking for the local maximum value of pixel points and setting the gray value corresponding to the non-maximum point to 0, so that a large number of non-edge points can be eliminated. The result of the image after NMS processing is shown in Figure 3c.

#### 2.2.4. Double Threshold Detection and Edge Connection

After non-maximum suppression, there are still many possible edge points; therefore, a double threshold was set up for further detection and edge connectivity, assuming two kinds of edges; after non-maximum suppression of the edge point, gradient values exceeding T1 are called strong edges, gradient values less than T1 but greater than T2 are called weak edges, and gradient values less than T2 are not considered edges. When T1 is too large, false edges are avoided to be extracted. However, due to the high threshold, the edge of the generated image may not be closed, so T2 is introduced as a low threshold. In an image with a high threshold, edges are linked into contours. When the end of the contour is reached, the algorithm will look for the point that meets the low threshold in the 8 neighborhood points of the breakpoint, and then collect new edges according to this point, until the image edge is closed. Usually, the effect is better when the ratio of T1 to T2 is 2:1, and in this paper, the value of T1 is 110 and the value of T2 is 55. The final result of the image processing is shown in Figure 3d.

### 2.3. Hyperspectral Crack Detection Principle and Classification Method Based on EfficientNet

#### 2.3.1. Principle of Crack Detection Based on Spectral Information

In related studies, the method of crack detection using hyperspectral technology was mainly applied to achieve the target of classification of detected substances by detecting the different properties of spectral absorption and reflection of different substances [22]. Specifically, this can be reflected in the hyperspectral spectrum. Different substances have different spectral information, and there are significant differences between substances in terms of their characteristic bands. The essence of detection or the change in reflectance varies. Detecting whether there are cracks or defects on the surface of a material also considers the observation of reflectivity; when incident light hits the surface of an object, some of the light is absorbed by the material, whereas the remainder is reflected in a specular and diffuse manner according to the surface roughness of the object, thus causing scattering. 

A hyperspectral imager can be placed in the vertical direction of the target object to be measured, whereby the hyperspectral image of the target is extracted vertically. There is a certain correlation between the surface state of an object and the scattering intensity distribution. Fewer surface cracks and defects lead to a stronger reflection and weaker scattering, and vice versa, as shown in Figure 4. At this time, the scattered light intensity generated by the same object on the hyperspectral imager is different due to the different surface states, thus producing different hyperspectral images. The spectral information at the crack location and the spectral information at the normal location differ to some extent, and the normal state and the crack state can be distinguished according to this spectral difference combined with a classification algorithm.

#### 2.3.2. Pattern Recognition of Porcelain Insulator Surface State Based on EfficientNet

Image diagnosis based on computer vision technology is limited by its detection principle, and the surface contamination of insulators can affect the accuracy of judging cracks. At this point, the insulator states can be distinguished by spectral analysis, and further classification of insulator states can be carried out according to the principle of the different spectra formed following the reflection of different substances. A deep learning method can be applied to identify and classify insulator states.

In the deep learning of images, width, depth, and resolution are the main factors affecting the prediction accuracy of neural networks. EfficientNet considers not only the adjustment of a single dimension [23,24,25] but also the comprehensive expansion of width, depth, and resolution. It also balances the forecast accuracy and resource footprint of the network. Therefore, EfficientNet was selected to classify the data. 

In the process of expansion, in addition to predicting accuracy, the parameters to be considered are the calculated number of floating-point operations per second (FLOPS)and the number of parameters to better measure the footprint of the convolutional neural network. Optimization of these parameters can reduce the neural network’s optimization search space.
(5)$$\underset{d,w,r}{\max}\;Accuracy\left(Net\left(d,w,r\right)\right),$$
(6)$$s.t.Net\left(d,w,r\right)=\underset{i=1\cdots s}{⊙}f_{i}^{d\cdot {\hat{L}}_{i}}\left(X_{\langle r\cdot {\hat{H}}_{i},r\cdot {\hat{W}}_{i},\omega \cdot {\hat{C}}_{i}\rangle }\right),$$
where EfficientNet is expressed as Net, and d,w,and r are the depth, width, and resolution, respectively. The constraint conditions stipulate the scope of network expansion and contraction.
(7)memory(Net)≤maximum limits_memory
(8)FLOPS(Net)≤maximum limits_FLOPS

When maximizing the training accuracy, we constrain the number of parameters in terms of the convolutional neural network: memory and FLOPS.

Changes between different dimensions are also taken into account, such as for images that have been expanded to a high resolution, with the need to increase the depth of the network to aid in capturing features with more detail. The following concept of composite coefficients is ϕ, therefore, proposed to explain the rule-based scaling of the entire network in terms of width, depth, and resolution:(9)$$\begin{array}{c} \left\{ \begin{array}{l} depth:d=\alpha^{\phi }\\ width:w=\beta^{\phi }\\ resolution:r=\gamma^{\phi } \end{array},\right.\\ s.t.\;\alpha \cdot \beta^{2}\cdot \gamma^{2}\approx 2,\alpha \geq 1,\beta \geq 1,\gamma \geq 1 \end{array}$$
where α, β, and γ are constants, which represent the initial values of depth, width, and resolution, respectively, the initial values are 1. α, β, and γ can be adjusted in a relatively small range under the constraints of the compound coefficients. As a compound coefficient, ϕ determines how many more resources are available for computing in the process of network expansion. Assuming that we use 2N times the computational resources, we can simply increase the depth of the network by αN times, the width by βN times, and the image size by γN times, where α, β, and γ are the constant coefficients determined by the tiny grid search on the original small model. Then, fixed ϕ=1, assuming that there are at least twice as many resources available. Then, α, β, and γ can be searched in a small range for network structure optimization. After calculation, we found that α=1.2, β=1.1, and γ=1.15 represent the optimal set of solutions. 

The resulting EfficientNet consisted of Stem, 16 Blocks, Con2Ds, Global Average, Pooling2D, and Dense, with 16 Blocks at its core. The network structure of the baseline network EfficientNet-B0 is shown in Figure 5.

### 2.4. Crack Positioning and Identification Model of Porcelain Insulators Based on Hyperspectral Data

The advantage of hyperspectral data in detecting insulator cracks is that they can make full use of image and spectral information to precisely and accurately locate and identify insulator cracks; therefore, this paper proposes a model for the positioning and identification of insulator cracks on the basis of map binding. The framework for the entire model is shown in Figure 6.

## 3. Experiment and Data Processing

### 3.1. Hyperspectral Detection Platform and Experimental Process

The laboratory hyperspectral image acquisition system consisted mainly of a hyperspectral imager, two 800 W halogen lamps, a sampling table, a stepper motor control system, a calibration white board, a computer, and a software platform, as shown in Figure 7. The hyperspectral camera was SPECIM FX10, which can obtain 400–1000 nm reflectivity intensity information for a total of 224 bands in the coverage range; the main specifications of the hyperspectral camera are shown in Table 1.

The two 800 W halogen tungsten lamps provided uniform lighting in the darkroom. The stepper motor control system controlled the movement of the hyperspectral camera along the *Z*-axis and the movement of the sampling table along the *X*-axis. The calibration white board was a standard Polytetrafluoroethylene (PTFE) white board with a reflectivity of 99%; that is, the default assumption is that it does not absorb light at all, completely reflects the waveband within the imaging range, and is collected as a white calibration image. The image is then captured with a reflectivity of 0% when the lens cover is closed as a dark correction image, which is then used for black-and-white correction of the image. The computer and software platform enabled the acquisition and processing of hyperspectral data.

The sample selected some porcelain insulators replaced from the field and some new porcelain insulators as experimental samples. The experiment consisted of several steps. Firstly, the hyperspectral camera lens was adjusted downward from the sample by 90 cm, with the two 800 W tungsten halogen lamps at an angle of 45° distributed symmetrically on both sides of the sample. Secondly, the camera cover was closed to capture the dark correction image. Thirdly, the hyperspectral camera captured the white board information as a white calibration image. Fourthly, the sample was placed on the sampling table, which was then adjusted along the *x*-axis using the motor control system to achieve a hyperspectral sweep of the sample, before transmitting the collected data to the computer for further processing. Lastly, the sample was replaced, the above steps were repeated.

Following the collection of hyperspectral data, the image and spectral data of the porcelain insulator sample could be obtained, and the information contained in the sample could be represented as a three-dimensional spectral data graph, as shown in Figure 8.

### 3.2. Hyperspectral Data Processing

In the process of collecting hyperspectral image data, there may be uneven light distribution, leading to image noise and dark currents in bands with a weak light intensity distribution. Therefore, it is often necessary to preprocess the original image obtained in the experiment after the completion of image collection to facilitate the subsequent analysis of hyperspectral data [26,27].

#### 3.2.1. Black-and-White Correction

Before the collection of hyperspectral image data, in order to overcome the influence of image noise and dark currents in the band with a weak light intensity distribution, a sample hyperspectral image was first collected by scanning the standard white board to collect all white calibration images with 99% reflectivity; then, the lens cap was covered to collect all black calibration images with 0% reflectivity. The reflectivity correction of the collected hyperspectral data could then be realized through black-and-white correction. The correction algorithm was as follows:(10)$$R=\frac{S-D}{W-D},$$
where *S* represents the original spectral image, *D* represents the all-black calibration image, *W* represents the all-white calibration image, and *R* represents the black-and-white-corrected image data. The sample spectral lines corrected using the black and white images are shown in Figure 9a,c, displaying a few burrs and basically smooth spectral lines.

#### 3.2.2. Multiplicative Scatter Correction

Multiplicative scatter correction (MSC) is a common processing method in multi-wavelength calibration modeling. MSC can effectively eliminate the spectral differences caused by different scattering levels, thus enhancing the correlation between spectrum and data. This method corrects the baseline shift and deviation of spectral data through the ideal spectrum. However, in practice, we cannot obtain the true ideal spectral data; thus, we often assume the average value of all spectral data as the ideal spectrum, obtained as follows:(11)$$\bar{Data}=\frac{\sum_{i=1}^{n}Data_{ij}}{n}.$$

The spectrum of each sample can then be regressed in a unit linear regression with the average spectrum, and the problem of least squares is solved to obtain the baseline translation and offset of each sample.
(12)$$Data_{i}=k_{i}\bar{Data}+b_{i}$$

The spectrum of each sample is then corrected to obtain the corrected spectrum.
(13)$$Data_{i\left(MSC\right)}=\frac{Data_{i}-b_{i}}{k_{i}}$$

The MSC-corrected spectrum is shown in Figure 9b,d, where it can be seen that the hyperspectral data processed by the MSC method effectively eliminated the error of the spectral data due to scattering level. Furthermore, the baseline translation and the offset of each spectrum were corrected under the reference of the standard spectrum, the spectral absorption information corresponding to the sample composition content was not affected during the data processing process, and the signal-to-noise ratio of the spectrum was improved.

### 3.3. Wavelength Dimension Reduction Based on End-Member Extraction 

Hyperspectral detection obtains too much multi-wavelength spectral information, and some of these bands do not contain important features that explain the results [28,29]. Therefore, in order to improve the recognition rate, classification accuracy, and algorithm operation speed of hyperspectral images, it is necessary to select a band and reduce its dimensionality, facilitating the understanding and analysis of the data.

#### 3.3.1. Minimum Noise Fraction (MNF) Transformation

MNF transformation was used to determine the inherent dimensions of image data (i.e., the number of bands) [30,31], separate the noise in the data, and reduce the computational demand in subsequent processing. MNF transformation essentially involves two cascading principal component transformations. The first transformation, based on the estimated noise covariance matrix, is used to separate and readjust the noise in the data. This operation results in the transformed noise data with minimal variance and no correlation between bands. The second step is the standard principal component transformation of noise-whitened data. MNF transformation, like principal component analysis (PCA), is an orthometric transformation, whereby the elements in the vector obtained after the transformation are not related to each other. The first component concentrates a large amount of information, where an increase in dimensionality leads to the image quality gradually decreasing, according to the signal-to-noise ratio. This allows overcoming the impact of noise on image quality. MNF transformation is superior to PCA transformation because the noise during the transformation has unit variances and is not related between bands.

#### 3.3.2. Pure Pixel Index (PPI) extraction 

When all elements in the spectral feature space are projected onto a unit vector u, the end elements are projected to both sides of u, and the mixed elements are projected to the middle [32]. Therefore, the image can be projected on n random unit vectors, and the number of times each unit is projected to the endpoint can be recorded, i.e., the pure unit index. A greater number of projections increases the probability of the cell being pure. As shown in Figure 10, $u_{1}$, $u_{2}$, and $u_{3}$ are three random unit vectors, with black dots representing pixels distributed in feature space, and A, B, C, and D represent the four-point pure-image index of 2,2,1,1 [33].

An increase in the number of extracted endmembers results in them having less influence on the structure of the hyperspectral image in the band space, i.e., a few endmembers in the front can constitute the general framework of the whole hyperspectral image in the band space, while the endmembers in the back contribute little to the framework [34]. As shown in Figure 11, in this paper, when the number of endmembers was 10, the general framework of the whole hyper spectrum was basically formed, i.e., a dimensionality reduction of the band was completed. The dimension reduction after extraction resulted in the bands of 403, 543, 599, 620, 653, 669, 680, 710, 803, and 998.

## 4. Results and Analysis

### 4.1. Results of Crack Localization of Porcelain Insulator

In this paper, two sample examples were used to evaluate the crack location. The process of crack location is shown in Figure 12a–d and Figure 13a–d.

The crack location process based on computer vision is the same as that in Section 2.2. Firstly, the original color image processed by grayscale image is sorted out, and Gaussian denoising is carried out to obtain effective data that can be used for edge recognition (see Figure 12a and Figure 13a), which can be used for edge recognition (see Figure 12b and Figure 13b). In order to effectively restrain the width of the edge and realize the accurate connection of the edge, in this paper, NMS Method and Double Threshold Detection Method are used to further process the edge. The final crack location identification results can be obtained (see Figure 12d and Figure 13d). (Some of the circular edges identified in the image are caused by light reflecting off the surface of the porcelain).

In the process of the crack location, some factors of the porcelain insulator greatly influence the outcome, such as image edge recognition accuracy, the influence of noise, and the edge width control. In this paper, the methods of denoising and edge width reduction were applied to extract a more accurate crack location. This method of crack identification and location through image edge processing is a universal method for crack identification problems. However, it needs to replace the appropriate denoising method and gray calculation method according to the different characteristics of the image and adjust the parameters T1 and T2 in double threshold detection.

### 4.2. Classification Results of the Surface State of Porcelain Insulators

After obtaining the surface state position of the crack from the previous step, the effective spectral information of the sample was selected to calculate the spectral mean of its region of interest (ROI), which was then plotted against the wavelength to construct a spectral map of the region used for state classification. The spectra of four sample regions were selected, whereby Figure 14a,b show the maximum, minimum, and mean values of the ROI of the crack sample, while Figure 14c,d show the maximum, minimum, and mean values of the ROI of the normal sample.

On the basis of obtaining certain defect samples and considering the dependence of convolutional neural network on data samples, the number of available training samples was finally obtained by means of data enhancement and data generation. A total of 1200 groups of porcelain insulator samples were collected, including 600 normal samples and 600 crack samples. Due to the relatively small amount of dataset, it was divided into a training set and a test set in a ratio of 7:3. We trained our model with Pytorch 1.8.0 on a machine equipped with a GeForce RTX 1080Ti graphics processing unit (GPU), an Intel i7-8700 central processing unit (CPU), and 16 GB of random-access memory (RAM). As the other model code, we used Pycharm and Anaconda, which are based on Python 3.7. The batch size during model training was 32, the learning rate was 0.01, the momentum was 0.9, and the number of epochs was 100.

To understand the generalization capability of the model, cross-validation is introduced. Therefore, when we divide a data set, we usually divide the data set into three data sets. The three data sets are training set, validation set, and testing set. In the training process, the K-fold cross-validation method is adopted in this paper to increase the utilization rate of data and prove the generalization performance of the model to a certain extent. In this paper, five-fold cross-validation is adopted. The schematic diagram of five-fold cross-validation is shown in Figure 15. The steps are as follows: (1) Four folds of train data (Train) and one fold of validation (Val) are separated from the training set; Train for training and Val for validation. (2) One of the five copies was successively selected as the Val dataset for model training and validation, so there are five combinations of training sets and validation sets, named Model1-5, respectively. (3) Each model was trained three times and the accuracy was averaged; then, selected the trained model with the largest average accuracy on Val as ‘best model’. (4) Evaluate ‘best model’ in the testing Set.

The result of the five-fold Cross-Validation is shown in Table 2. Among them, the average value of the three times of verification accuracy of Model4 is 93.13%, which is higher than that of Model1 (92.92%), Model2 (90.42%), Model3 (88.96%), and Model5 (89.38%), so Model4 is selected as the ‘best model’ to evaluate the model on the testing set.

To verify the performance of the proposed approach, typical methods in the field of machine learning and deep learning (support vector machine (SVM), decision tree (DT), random forest (RF), Back Propagation Neural Networks (BPNN), and AlexNet) were used to compare and classify the results related to porcelain insulator state and validation. For the traditional machine learning methods, the max value, the min value, root-mean-square deviation, standard deviation, skewness, and peak-to-peak value were extracted as the feature parameters. For EfficientNet, we directly identified and classified the average spectra, whereby the sample images were all drawn within the unified coordinate range, and the size and resolution of the images sent to the model after normalization were also the same.

As shown in Figure 16, the overall recognition accuracy of EfficientNet was 96.9%, markedly higher than the results obtained for SVM, decision tree, random forest, BPNN, and AlexNet (82.2%, 58.9%, 76.7%, 81.1%, and 91.7%, respectively). This indicates that EfficientNet can effectively distinguish each defect type; in fact, its recognition rate for the two states reached 95.6% and 98.3%. 

Moreover, according to the recognition results, the models had different recognition performances for each type of defect. As shown in Figure 16, for the decision tree, the recognition accuracy of the normal sample (61.1%) was higher than that of the crack area (56.7%); in addition to the decision tree model, the accuracy of other models on crack samples is little higher than that on normal samples. This phenomenon may have something to do with sample collection and sample labeling. In the data set, the maximum value of the samples at the crack location is concentrated in a small range, which is around 800. For normal samples, some samples with surface contamination during the collection process had also be included in the data set. Compared with crack samples, such samples were also labeled as normal samples in the binary classification model in this paper; these samples with surface contamination will result in a maximum spectral range of 800 to 1500 for normal samples. Therefore, in the process of pattern recognition, it is more possible to mistake the normal sample for the crack sample.

The decision tree had the lowest overall recognition rate of 58.9%, which is because it is easy to produce an overfitting phenomenon in the process of learning, and the feature extracted by hand may not cover all the information of the main features of the image, whereas the random forest model based on the decision tree adds an integrated learning method to improve the accuracy of the recognition result (76.7%). EfficientNet showed the best performance among these models, with the recognition accuracy still at a very high level of 96.9%, and the advantages of network structure can be shown in the results. The expansion of the network in multi-dimensional effectively improves the accuracy of pattern recognition. 

## 5. Conclusions

In this paper, we proposed a porcelain insulator detection method based on hyperspectral detection technology, which can satisfy the detection requirements for online and noncontact detection. It works by utilizing the hyperspectral measurement process of image data and spectral data, using the resulting information to distinguish between normal and crack states, thus providing an assessment of the crack condition. The main conclusions of this paper are provided below: 

(1) The porcelain insulator detection method based on hyperspectral detection technology proposed in this paper can realize the noncontact detection of porcelain insulator cracks. This method is an important supplement to the existing crack detection methods for porcelain insulators; thus, it has good application prospects for potential fault detection and online monitoring of insulators, and it conforms to the current trend of online monitoring of power equipment. 

(2) Through image processing steps such as edge detection, the detection model proposed in this paper can accurately locate cracks and suspected cracks on the surface of porcelain insulators.

(3) EfficientNet has effectively implemented porcelain insulator surface state pattern recognition. The surface state of porcelain insulator samples can be classified using EfficientNet with 96.9% accuracy, higher than the results of SVM, DT, RF, BPNN, and AlexNet (82.2%, 58.9%, 76.7%, 81.1%, and 91.7%, respectively). It is also proved that the combination of deep learning method and hyperspectral data can achieve the desired effect in the pattern recognition of porcelain insulator surface state. Its capabilities of online monitoring and fault diagnosis of electric power equipment have broad application prospects.

## Figures and Tables

**Figure 1 entropy-23-00486-f001:**
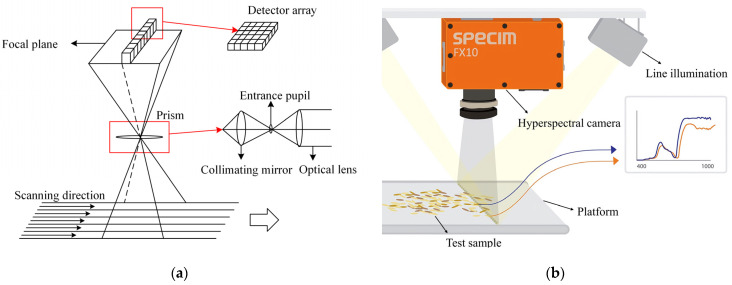
Crack location diagram: (**a**) structure of hyperspectral camera; (**b**) schematic diagram of hyperspectral imaging principle.

**Figure 2 entropy-23-00486-f002:**
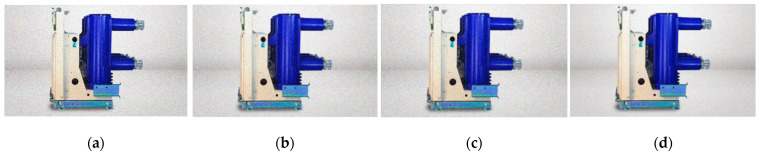
Comparison of filtering methods: (**a**) image after adding Gaussian noise; (**b**) image after mean filtering; (**c**) image after box filtering; (**d**) image after Gaussian filtering.

**Figure 3 entropy-23-00486-f003:**
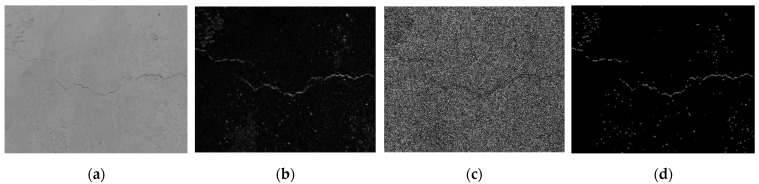
Crack location diagram: (**a**) image after Gaussian filtering; (**b**) gradient amplitude map; (**c**) image after Non-Maximum Suppression(NMS) processing; (**d**) image after double threshold detection and edge connection.

**Figure 4 entropy-23-00486-f004:**
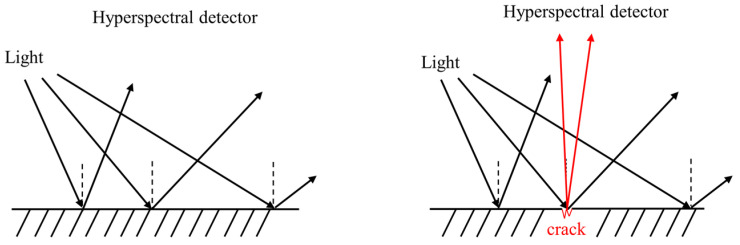
The principle of crack detection based on the hyperspectral graph.

**Figure 5 entropy-23-00486-f005:**
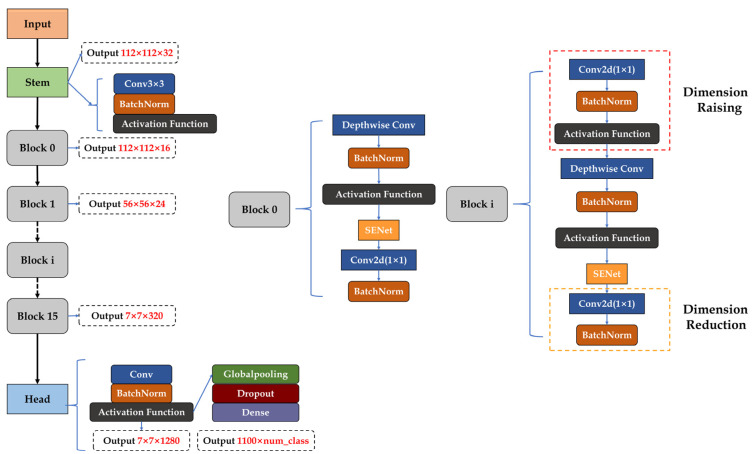
EfficientNet-B0 network architecture.

**Figure 6 entropy-23-00486-f006:**
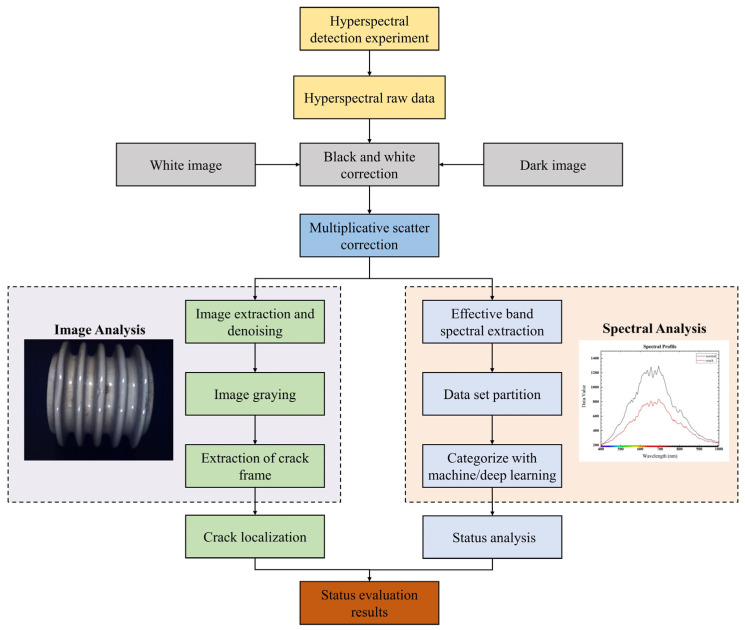
The overall framework of the insulator crack positioning and identification model.

**Figure 7 entropy-23-00486-f007:**
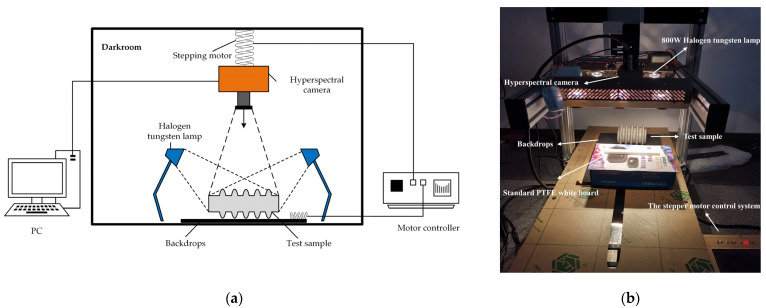
Hyperspectral detection platform: (**a**) diagram of the hyperspectral detection platform; (**b**) hyperspectral detection experimental platform.

**Figure 8 entropy-23-00486-f008:**
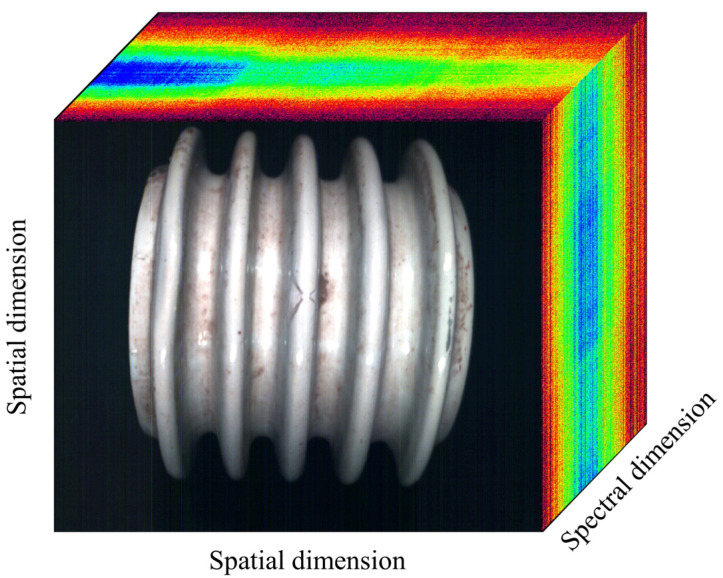
Hyperspectral three-dimensional (3D) data map of insulator.

**Figure 9 entropy-23-00486-f009:**
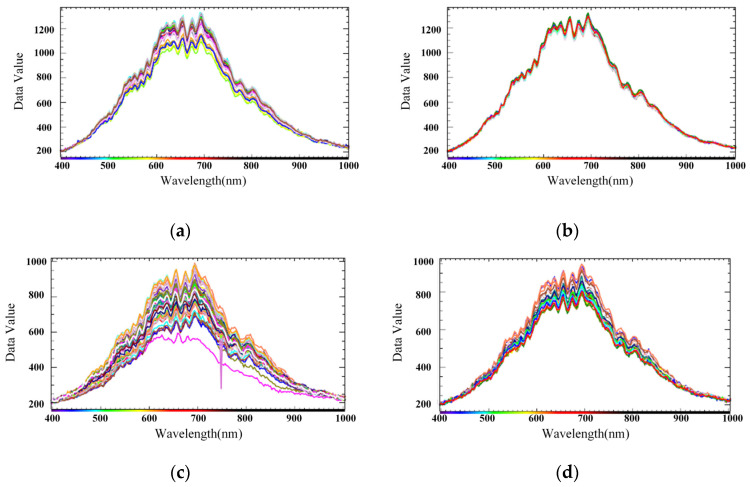
Black-and-white correction and multiplicative scatter correction (MSC): (**a**) normal sample spectrum after black-and-white correction; (**b**) normal sample spectrum after MSC; (**c**) spectrum of crack samples after black-and-white correction; (**d**) crack sample spectrum after correction.

**Figure 10 entropy-23-00486-f010:**
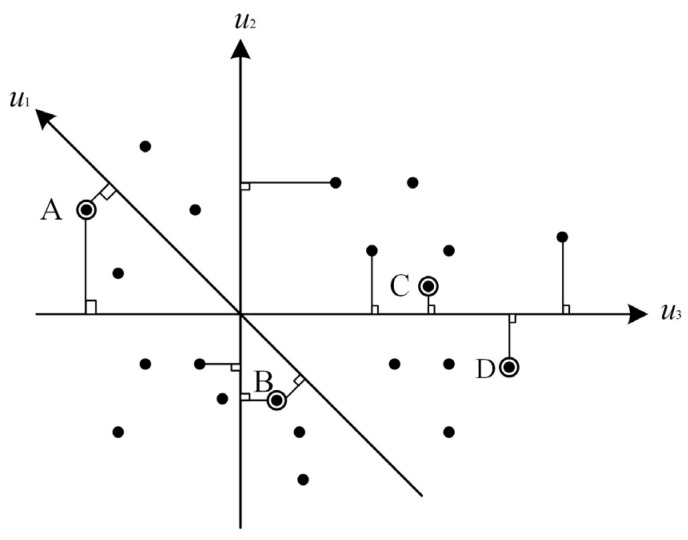
Schematic of pure pixel exponential end-member extraction algorithm.

**Figure 11 entropy-23-00486-f011:**
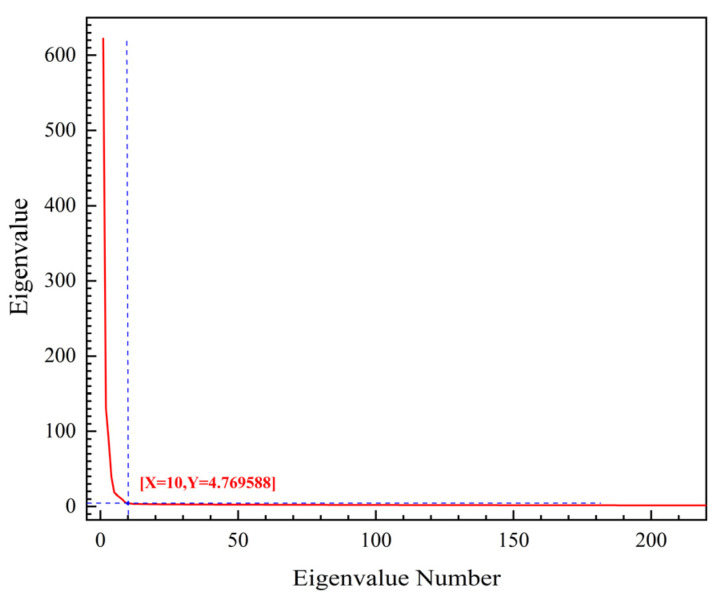
Spectral dimension reduction results.

**Figure 12 entropy-23-00486-f012:**
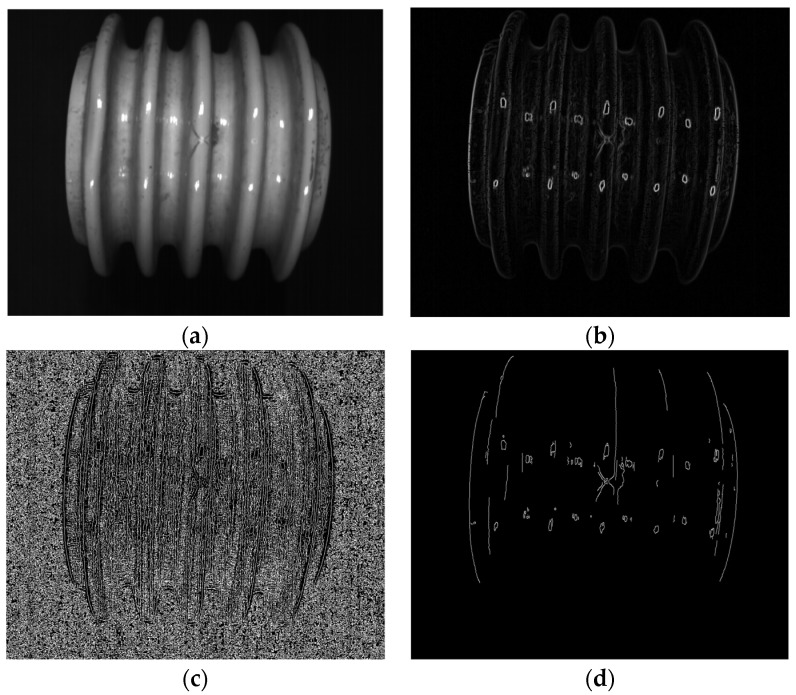
Crack localization of porcelain insulator sample 1: (**a**) image after Gaussian filtering; (**b**) gradient amplitude map; (**c**) image after NMS processing; (**d**) image after double threshold detection and edge connection.

**Figure 13 entropy-23-00486-f013:**
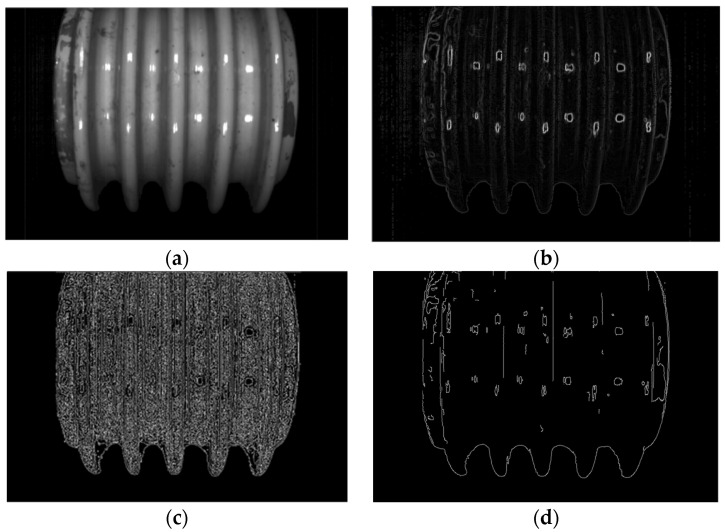
Crack localization of porcelain insulator sample 2: (**a**) image after Gaussian filtering; (**b**) gradient amplitude map; (**c**) image after NMS processing; (**d**) image after double threshold detection and edge connection.

**Figure 14 entropy-23-00486-f014:**
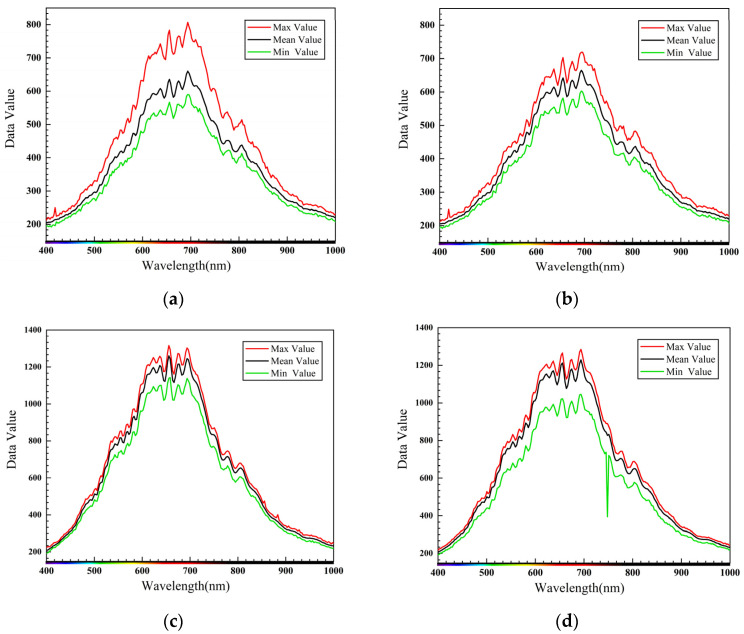
Spectral sample collection: (**a**) crack region spectrum of sample 1; (**b**) crack region spectrum of sample 2; (**c**) normal region spectrum of sample 3; (**d**) normal region spectrum of sample 4.

**Figure 15 entropy-23-00486-f015:**
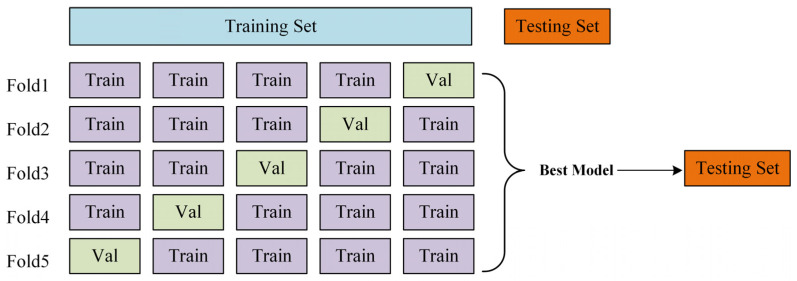
The schematic diagram of 5-fold cross-validation.

**Figure 16 entropy-23-00486-f016:**
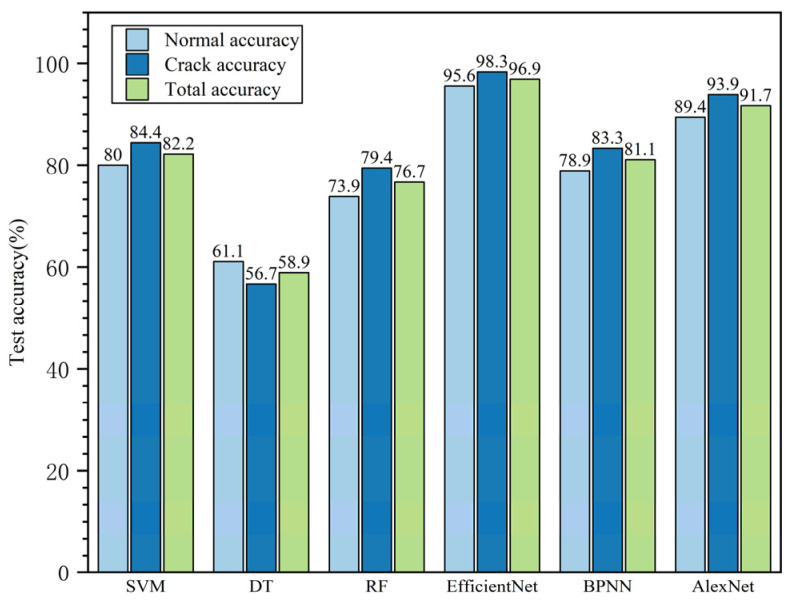
Porcelain insulator state identification accuracy results. SVM, support vector machine; DT, decision tree; RF, random forest; BPNN, back propagation neural networks.

**Table 1 entropy-23-00486-t001:** The main parameters of the hyperspectral camera.

Main Parameter	Meaning	Value
Spectral range(nm)	Measurable range	400–1000
Spectral band	Spectral range divided into intervals	224
Spectral resolution	Recording width in the wavelength direction	5.5 nm
Signal noise ratio (SNR)	A higher SNR value denotes lower noise	600:1
Charge-coupled device (CCD) pixel	Affects low illumination and noise	1392 × 1040

**Table 2 entropy-23-00486-t002:** Result of 5-fold Cross-Validation. Acc, accuracy.

	Model 1	Model 2	Model 3	Model 4	Model 5
First Validation Acc (%)	85.63	95.00	95.00	96.88	82.50
Second Validation Acc (%)	95.00	78.12	88.13	95.63	93.75
Third Validation Acc (%)	98.13	98.13	83.75	86.88	91.88
Average Acc (%)	92.92	90.42	88.96	93.13	89.38

## Data Availability

The data presented in this study are available on request from the corresponding author.

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
