# Peer review of "Porcelain Insulator Crack Location and Surface States Pattern Recognition Based on Hyperspectral Technology"

_entropy, 2021, doi:10.3390/e23040486_

Round 1

Reviewer 1 Report

In the paper, an approach for porcelain insulator crack detection is proposed. The method is simple and efficient. However, the following issues should be addressed:
1.    The contributions of the study should be enlisted at the end of Section I.
2.    The association of the study with entropy is not revealed. The scope of the journal should be better addressed.
3.    The insulation requirements should be introduced. Which damages or crack locations are considered relatively safe? The acceptable crack damage should be investigated. 
4.    The title suggests that cracks are going to be recognized and their location determined. However, the classification results reveal that a binary classification is performed, differentiating samples into damaged or non-damaged categories. This should be explained as finding the location is a different from the recognition (just think of the face detection and recognition approaches). If the location is not used in further stages of the insulator processing (see point 3), this should be clarified. Note that the accuracy of the crack location is not discussed here.   
5.    The mentioned higher safety coefficients (Abstract) should be explained. Alternatively, please rewrite the paragraph if it is a matter of justification of the approach. 
6.    The paper lacks a discussion on the influence of parameters on the performance of the method. 
a.    Show the results for at least three other image denoising approaches. The used Gaussian filter is often outperformed by new approaches in this area. Hence, its application should be justified.
b.    There many methods that can be used for gradient calculation. Show the results with e.g., Prewitt mask instead of the Sobel.
c.    The influence of T1 and T2 is not investigated (introduced in Section 2.2.3)
d.    The influence of alpha, beta, and gamma is not discussed. Their values are arbitrary.
e.    Please show the results for other examples of the network. The AlexNet or Inception should be compared here.
7.    The data division into training-testing subsets is arbitrary. Here, 5-fold cross-validation should be used along with reports, in Table 2, containing accuracies from each fold. 
8.    Fig. 15 contains some cross-validation results. It is not coherent with used data –division and results in Table 2.  
9.    Can the data augmentation be used to improve the results of the EfficientNet? This would address the claim that the accumulation of data sample can expand the efficiency of the network (lines 400 and 424). This claim is not supported experimentally. 
10.    The reported experiments cannot be replicated. Please share the source code with readers, ensuring the reproducibility of the results (mandatory). The samples used in experiments (the dataset) should be made publicly available. This would greatly promote the paper.
11.    Minor comments:
a. The paper requires proofreading (e.g., Eq. (2) should be better formatted). 
b.    The important parts in Figs. 1-2 are not described in Section 2.1. They can be merged into Fig. 1.a and Fig. 1.b.  
c.    Accuracy is measured in percent (please correct Fig. 15.)

Author Response

Thank you for your comments.,my point-to-point response has been compiled into PDF and attached.Please see the attachment.

Reviewer 2 Report

Proposed manuscript deals with a very intersting topic of porcelain insulator crack localisation using advanced image processing methods (mainly edge detection and patern recognition). The motivation is essential: porcelain insulation, play an important role in ensuring that electrical equipment meets insulation requirements. Authors introduce the detector based on hyperspectral imaging.

Here I have to state that the manuscript is almost camera-ready in present form because I have no major comments.

I have two minor comments:

1. professional proof-reading should be done
2. From the figure 12 and texts in section Results it is not clear what alogrithm is used for removing the "essential" edges (made by the construction of the insulator).

In general, I can see a great potential in present  form of the manuscript therefore I recommend to accept the manuscript after minor revision.

Author Response

Thank you for your comments,my point-to-point response has been compiled into PDF and attached. Please see the attachment.

Round 2

Reviewer 1 Report

The revised manuscript addresses all identified issues.

Author Response

We thank the reviewer for his time and his very positive and constructive remarks,

which helped to improve the quality and clarity of the manuscript.